# Halotolerant Microbial Consortia for Sustainable Mitigation of Salinity Stress, Growth Promotion, and Mineral Uptake in Tomato Plants and Soil Nutrient Enrichment

Chintan Kapadia [1], R. Z. Sayyed [2,*], Hesham Ali El Enshasy [3,4,*], Harihar Vaidya [5], Deepshika Sharma [6], Nafisa Patel [6,*], Roslinda Abd Malek [3], Asad Syed [7], Abdallah M. Elgorban [7], Khurshid Ahmad [8] and Ali Tan Kee Zuan [9,*]

1   Department of Plant Molecular Biology and Biotechnology, ASPEE College of Horticulture and Forestry, Navsari Agricultural University, Navsari 396450, India; chintan_bt@yahoo.co.in
2   Department of Microbiology, PSGVP Mandal's Arts, Science, Commerce College, Shahada 425409, India
3   Institute of Bioproduct Development (IBD), Universiti Teknologi Malaysia (UTM), Johor Bahru 81310, Malaysia; roslinda@ibd.utm.my
4   City of Scientific Research and Technology Applications (SRTA), New Burg Al Arab, Alexandria 21934, Egypt
5   Soil and Water Management Research Unit, Navsari Agricultural University, Navsari 396450, India; hariharbvaidya@gmail.com
6   Naranlala College of Professional and Applied Science, Bhagwati Sankul, Eru Char Rasta, Navsari 396450, India; deepshika121095@gmail.com
7   Department of Botany and Microbiology, College of Science, King Saud University, P.O. Box 2455, Riyadh 11451, Saudi Arabia; assyed@ksu.edu.sa (A.S.); aelgorban@ksu.edu.sa (A.M.E.)
8   Department of Medical Biotechnology, Yeungnam University, Gyeongsan 38541, Korea; ahmadk@ynu.ac
9   Department of Land Management, Faculty of Agriculture, Universiti Putra Malaysia (UPM), Serdang 43400, Malaysia
*   Correspondence: sayyedrz@gmail.com (R.Z.S.); henshasy@ibd.utm.my (H.A.E.E.); nafisa.z.patel@gmail.com (N.P.); tkz@upm.edu.my (A.T.K.Z.)

**Abstract:** Salinity significantly impacts the growth, development, and reproductive biology of various crops such as vegetables. The cultivable area is reduced due to the accumulation of salts and chemicals currently in use and is not amenable to a large extent to avoid such abiotic stress factors. The addition of microbes enriches the soil without any adverse effects. The effects of microbial consortia comprising *Bacillus* sp., *Delftia* sp., *Enterobacter* sp., *Achromobacter* sp., was evaluated on the growth and mineral uptake in tomatoes (*Solanum Lycopersicum* L.) under salt stress and normal soil conditions. Salinity treatments comprising Ec 0, 2, 5, and 8 dS/m were established by mixing soil with seawater until the desired Ec was achieved. The seedlings were transplanted in the pots of the respective pH and were inoculated with microbial consortia. After sufficient growth, these seedlings were transplanted in soil seedling trays. The measurement of soil minerals such as Na, K, Ca, Mg, Cu, Mn, and pH and the Ec were evaluated and compared with the control 0 days, 15 days, and 35 days after inoculation. The results were found to be non-significant for the soil parameters. In the uninoculated seedlings' (control) seedling trays, salt treatment significantly affected leaf, shoot, root dry weight, shoot height, number of secondary roots, chlorophyll, and mineral contents. While bacterized seedlings sown under saline soil significantly increased leaf (105.17%), shoot (105.62%), root (109.06%) dry weight, leaf number (75.68%), shoot length (92.95%), root length (146.14%), secondary roots (91.23%), and chlorophyll content (−61.49%) as compared to the control (without consortia). The Na and K intake were higher even in the presence of the microbes, but the beneficial effect of the microbe helps plants sustain in the saline environment. The inoculation of microbial consortia produced more secondary roots, which accumulate more minerals and transport substances to the different parts of the plant; thus, it produced higher biomass and growth. Results of the present study revealed that the treatment with microbial consortia could alleviate the deleterious effects of salinity stress and improve the growth of tomato plants under salinity stress. Microbial consortia appear to be the best alternative and cost-effective and sustainable approach for managing soil salinity and improving plant growth under salt stress conditions.

**Keywords:** microbial consortia; PGPR; plant biomass; salinity stress; tomato

## 1. Introduction

Plants and their products serve as major sources of energy to humans; moreover, they have an important role in the sustainability of the ecosystem [1]. The soil along with the plants are adversely affected by the diverse abiotic and biotic factors. The resistant traits towards abiotic stress are generally complex and cannot be resolved by modifying a single factor. Soil salinity, or an excess amount of salt present in the soil, is one of the significant problems. Around one billion hectares of the world's agricultural land area is affected by salinity, and every year there is a 10% increase in saline land among which irrigated agricultural lands are severely affected. In India, seven million hectares of land are under salinity stress across the Indo-Gangetic plain [2], as well as other regions such as Gujarat, Rajasthan, and Madhya Pradesh in India. Salinity is a dominant abiotic stress that affects the productivity and quality of a crop [3–7]. A soil becomes saline when the electrical conductivity (Ec) in the soil surrounding the root zone rises to 4 dS/m (40 mM of NaCl) [4]. Soil salinity replaces the coagulators such as $Ca^{2+}$ and $Mg^{2+}$ ions, which are absorbed on the surface of the soil aggregates. Salinity adversely affects the majority of horticultural crops, especially vegetables as their tolerance level is 2 $dS/m^{-1}$ [1]. The gradual accumulation of salts would disturb the metabolic processes of the plants, and if such stress appears at the reproductive stage, then yield will be reduced to less than 50%. The salinity alters the water holding capacity of roots, which creates an accumulation of ions in the different compartments of the plant's tissue. The photosynthesis and movement of various compounds to diverse location is disturbed and as a result, the vegetative growth would be stalled. Scraping, flushing, and leaching are the mechanical means of the recla-mation of saline soil and involve higher cost and management skills [8]. Whereas organic manures and gypsum could also be used to reduce the impact of salinity on crops [9]. It also affects the population and diversity of helpful soil micro-organisms, protozoan, and nematodes [5] and the biogeochemical processes carried out by these organisms [6]. Salinity negatively impacts the growth, yield, and physiological attributes in several plants, including tomatoes [7]. Various physical [8] and chemical [9] approaches have salvaged the saline soil. These methods are less effective, not eco-friendly, and not affordable for large-scale applications. In this regard, plant growth promoting rhizobacteria (PGPR) are suggested as the best alternative for saline soil reclamation. Plant growth promoting rhizobacteria (PGPR) belongs to the genera *Rhizobium*, *Bacillus*, *Klebsiella*, *Enterobacter*, *Azotobacter*, *Pseudomonas*, and *Streptomyces*, and have been previously reported by many researchers to improve the biomass of diverse crops in salinity stress [10]. The plant can withstand salt stress to a threshold value, and above that, it requires certain enzymes, phytohormones, osmolytes, solubilization of nutrients, and an increased absorption of competitive ions [11]. Numerous PGPR have been reported to provide various antioxidant enzymes and osmolytes [12]; however, the formulation of compatible yet different genera of microbes can significantly improve the growth of plants under saline conditions and may help in salinity amelioration [11–14]. During stress conditions, microbial consortia play multiple vital roles, such as plant growth promotion, osmoprotectant, antioxidant and biocontrol activity, and alleviating stress in soil [15]. More studies are needed to prove and establish the fact that a microbial consortium is better than the individual strains [16–18].

The present study was undertaken to evaluate the plant growth promotion effect of microbial consortia consisting of halophilic and halotolerant bacteria isolated from the saline environment on tomato plants under 0.2 Ec dS/m to 8 Ec dS/m containing soil.

## 2. Materials and Methods

### 2.1. Tomato Seeds and Microbial Culture

Tomato seeds (*Solanum lycopersicum* L. cv. Arka Vikas) were procured from the Indian Institute of Horticultural Research, *Bangalore* (IIHR) and sown in seedling trays filled with sterilized coco-peat under greenhouse conditions.

### 2.2. Microbial Consortia Preparation and Inoculation

The six strains of micro-organisms with different plant growth-promoting traits were evaluated in the laboratory and employed in this experiment (unpublished data, Table 1). Halo tolerant rhizobacterial strains were selected based on their phosphate solubilization and production of siderophores, ammonia, and indole acetic acid (IAA). The bacterial culture was grown in the 5 mL Luria–Bertani broth and kept in the shaking incubator for 10–12 h. 1 mL of culture from each strain was added to the 100 mL of Luria–Bertani broth and incubated for a further period. The colony-forming unit was checked every two hours to obtain $10^8$ colony-forming units per mL (CFU/mL). After transplanting, a 10 mL from this suspension was poured into the rhizosphere of the tomato seedling in the pots. There were two control, one received only water without micro-organisms, while the other with micro-organisms at 0.2 Ec dS/m.

**Table 1.** List of the microbes used in the consortia with their plant growth-promoting traits.

| Name of the Organisms | Accession Number | Ammonia Production | Phosphate Solubilization | IAA Production | Siderophore Production |
|---|---|---|---|---|---|
| *Achromobacter* sp. clone ADCNI | MH142385 | + | + | + | + |
| *Bacillus* sp. clone ADCNE | MH142391 | + | + | + | + |
| *Bacillus sonorensis* clone ADCNF | MH127536 | + | + | + | + |
| *Bacillus* sp. clone ADCNJ | MH142386 | + | + | + | + |
| *Delftia* sp. clone ADCNK | MH142388 | + | + | + | + |
| *Enterobacter* sp. clone ADCNP | MH142389 | + | + | + | + |

[+] = presence of trait.

### 2.3. Seedling Growth

Tomato seeds (*Solanum Lycopersicum L.* cv. Arka Vikas) were sown in seedling trays filled with sterilized coco-peat under greenhouse conditions at the Department of Plant Molecular Biology and Biotechnology laboratory, Navsari Agricultural University, Navsari, India. The seeds were irrigated with water, and fertigation was applied regularly using a commercial NPK: 6:12:36 (Sumukha Farm Products Pvt Ltd., Navsari, India) micronutrient solution. After emergence, the seedlings were allowed to grow in seedling trays for 25–30 days in the greenhouse under the control condition of 70% humidity and 30 °C temperature. The 25–30 individual seedlings were transplanted in the pot filled with soil and sand mixture (1:1) to obtain sufficient plants for various analyses. The different 0.2 Ec dS/m to 8 Ec dS/m Ec levels of the soil were maintained using sterilized seawater collected from the Dandi Sea, Navsari, Gujarat, India. This water was sterilized and mixed with bulk soil until the desired Ec was achieved, followed by filling in the respective pot. The number and name of the individual pots with the Ec level were attached to them. The plants were allowed to grow in the natural light and temperature.

Treatments

There were eight treatments as follows:

A1—Distilled water;

A2—Microbial consortia;

B1—Control Soil;

B2—2 Ec;

B3—5 Ec;

B4—8 Ec.



The combination of treatments would be A1B1, A1B2, A1B3, A1B4, A2B1, A2B2, A2B3, and A2B4. The experiment was conducted with three biological replications and three technical replications.

### 2.4. Measurement of Growth Parameters

After 0 days, 15 days, and 35 days of inoculation considered as C1, C2, and C3, respectively, five plants from each treatment were removed and subjected for analysis of plant growth parameters and soil mineral contents.

### 2.5. Measurement of Ec, pH, Mineral Content, and Chlorophyll Analysis

Soil pH and electric conductivity were measured by using the digital Ec meter according to the method of Corwin and Lesch [19]. Plant biomass and other parameters were measured by removing the intact plants from the pot and measuring the length of root and shoot, counting the number of leaves, and a number of secondary roots. Chlorophyll content was measured by using the acetone method [20]. Cation exchange capacity and exchangeable sodium percentage of the soil were estimated according to the method of Mehlich [21]. $Na^+$ and $K^+$ from the plants were extracted using the diacidic digestion method and estimated by a flame photometer [22]. Estimation of calcium and magnesium was performed by the ethylenediaminetetraacetic acid (EDTA) titrimetric method, while manganese and copper were determined using the Atomic Absorption Spectrophotometer (AAS) (Electronics Corporation of India Ltd. AAS 4141) [23].

### 2.6. Statistical Analysis

Data were statistically analyzed by three factorial analysis of variance (ANOVA) using the statistical programming language R [24] using the add-on 'agricolae' package [25], and the mean values were compared using Duncan's post hoc test to group them and assign letters depending on the difference between them. The significant differences were noted by the values of $p \leq 0.01$ and 0.05.

## 3. Results

### 3.1. Measurement of Plant Growth-Related Parameters

The inoculation of consortia and different Ec levels of the soil had a significant ($p < 0.01$) influence on the growth-related parameters of the tomatoes (Figure 1a–f). The plant growth-related parameters were adversely affected by the higher Ec levels of the soil. The plant growth-related parameters were studied at 0.2 dS/m to 8 dS/m under inoculated (uninoculated, A1) and consortia (A2) plants at 15 days (C1) and 35 days (C2). The shoot dry weight, root dry weight, and leaf dry weight reduced significantly by 63% (0.18 g), 57% (0.25 g), and 68% (0.27 g) when soil Ec was increased to 8.0 dS/m under the controlled uninoculated condition (Figure 1e,f). There was a substantial decrease in the number of growth parameters after 35 days of inoculation by 55% (0.44 g), 36.43% (0.44 g) and 49% (0.63 g) from their respective control at 8.0 dS/m (Table 2). The growth of the plant parts decreased due to saline stress, while the inoculated plants in treatments A2B1, A2B2, A2B3, and A2B4 showed significantly better performance compared to their uninoculated counterparts (Figure 1f). The inoculation of microbial consortia as A2 treatment to all B1 to B4 plants exhibited a decline in shoot dry weight, root dry weight, and leaf dry weight by 28.49% (0.52 g), 33% (0.36 g), and 39.60% (0.55 g) after 15 days, respectively, at 8.0 dS/m, while there was a reduction in the values by 26.65% (0.89 g), 22.39% (0.58 g), and 26.65% (1.13 g), respectively, at 35 days of post inoculation (Table 3) (Figure 1b,d,f). The inoculated plants showed higher values and lower reduction percentages for dry mass, which indicates the positive effects of the microbial consortia to the plants.

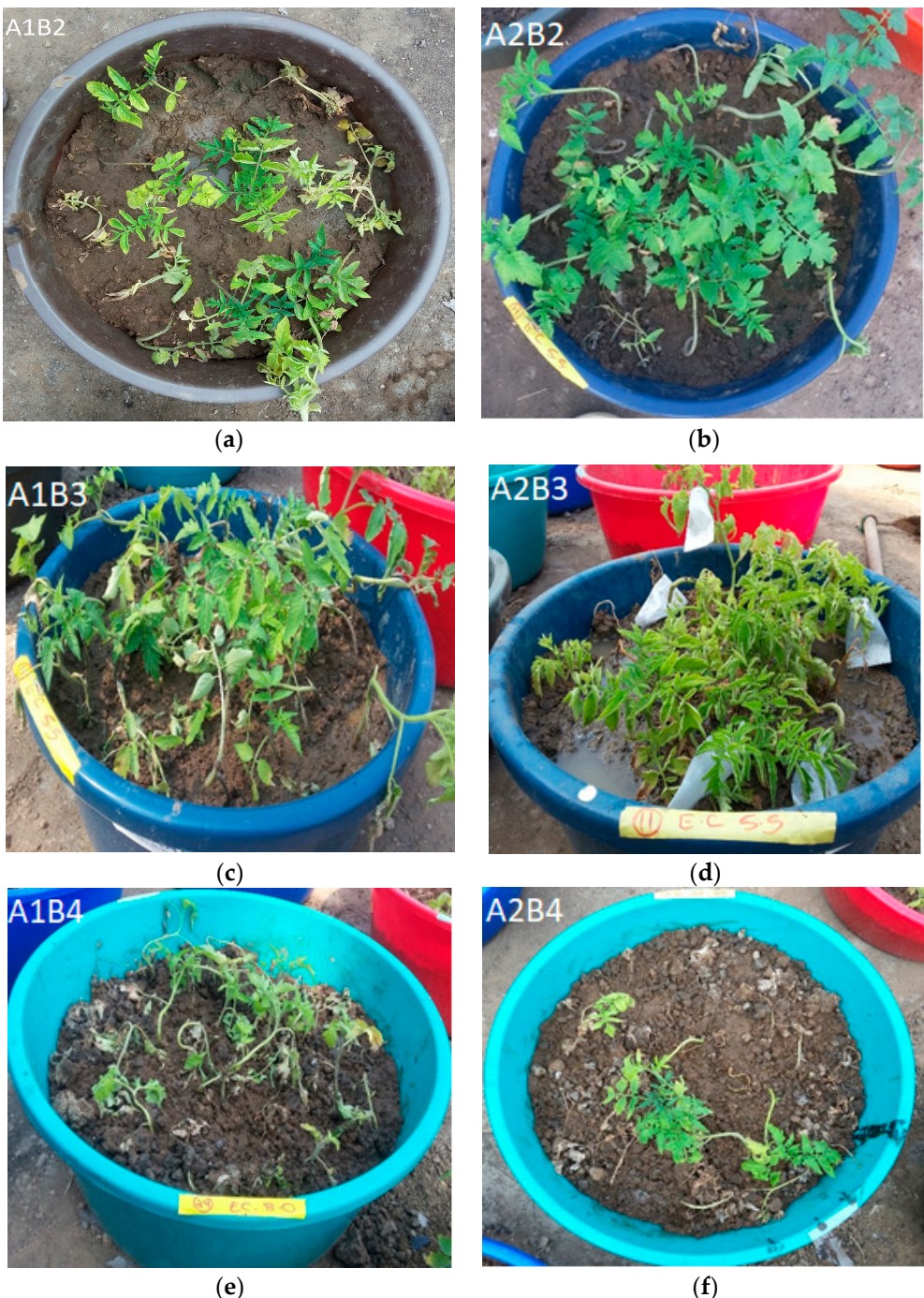

**Figure 1.** (**a**) Treatment A1B2—plant grown in uninoculated soil having an Ec value of 2.0. (**b**) Treatment A2B2—plant grown in soil having microbial consortia and an Ec value of 2.0. (**c**) Treatment A1B3—plant grown in uninoculated soil having an Ec value of 5.0. (**d**) Treatment A2B3—plant grown in soil having microbial consortia and an Ec value of 5.0. (**e**) Treatment A1B4—plant grown in uninoculated soil having an Ec value of 8.0. (**f**) Treatment A2B4—plant grown in soil inoculated with microbial consortia and having an Ec value of 8.0.

**Table 2.** The properties and mineral content of the soil during the experiment.

| Treatments | Ca (me/100 g) | Na (me/100 g) | K (me/100 g) | CEC (me/100 g) | ESP (%) |
|---|---|---|---|---|---|
| A1B1C1 | 45.23 [o] | 0.73 [o] | 0.24 [o] | 46.20 [o] | 1.59 [o] |
| A1B1C2 | 46.93 [o] | 0.90 [o] | 0.23 [o] | 48.06 [o] | 1.87 [o] |
| A1B1C3 | 43.50 [o] | 0.93 [o] | 0.25 [o] | 44.68 [o] | 2.08 [o] |
| A1B2C1 | 45.67 [l] | 5.34 [l] | 0.63 [l] | 51.63 [l] | 10.34 [l] |
| A1B2C2 | 41.63 [j] | 4.91 [j] | 0.55 [j] | 47.08 [j] | 10.42 [j] |
| A1B2C3 | 39.63 [hi] | 4.12 [hi] | 0.63 [hi] | 44.38 [hi] | 9.31 [hi] |
| A1B3C1 | 45.33 [g] | 7.39 [g] | 1.03 [g] | 53.76 [g] | 13.74 [g] |
| A1B3C2 | 44.90 [f] | 6.52 [f] | 1.21 [f] | 52.63 [f] | 12.40 [f] |
| A1B3C3 | 42.83 [f] | 5.58 [f] | 1.14 [f] | 49.56 [f] | 11.27 [f] |
| A1B4C1 | 41.17 [c] | 6.37 [c] | 0.43 [c] | 47.96 [c] | 13.28 [c] |
| A1B4C2 | 40.40 [b] | 5.97 [b] | 0.63 [b] | 47.01 [b] | 12.71 [b] |
| A1B4C3 | 38.42 [a] | 5.30 [a] | 0.55 [a] | 44.27 [a] | 11.98 [a] |
| A2B1C1 | 50.90 [n] | 1.17 [n] | 0.32 [n] | 52.39 [n] | 2.24 [n] |
| A2B1C2 | 50.40 [n] | 1.67 [n] | 0.54 [n] | 52.61 [n] | 3.19 [n] |
| A2B1C3 | 47.54 [n] | 2.13 [n] | 0.34 [n] | 50.01 [n] | 4.26 [n] |
| A2B2C1 | 52.00 [k] | 5.33 [k] | 0.54 [k] | 57.88 [k] | 9.23 [k] |
| A2B2C2 | 48.17 [l] | 4.03 [l] | 0.53 [l] | 52.73 [l] | 7.66 [l] |
| A2B2C3 | 46.60 [m] | 2.60 [m] | 0.61 [m] | 49.81 [m] | 5.23 [m] |
| A2B3C1 | 55.80 [h] | 6.92 [h] | 0.53 [h] | 63.25 [h] | 10.94 [h] |
| A2B3C2 | 45.79 [i] | 4.93 [i] | 0.73 [i] | 51.46 [i] | 9.59 [i] |
| A2B3C3 | 43.72 [j] | 3.84 [j] | 0.64 [j] | 48.20 [j] | 7.96 [j] |
| A2B4C1 | 47.00 [b] | 5.68 [b] | 0.64 [b] | 53.32 [b] | 10.66 [b] |
| A2B4C2 | 42.50 [d] | 4.09 [d] | 0.54 [d] | 47.14 [d] | 8.69 [d] |
| A2B4C3 | 40.69 [e] | 3.69 [e] | 0.64 [e] | 45.02 [e] | 8.19 [e] |
| A | ** | ** | ** | ** | ** |
| B | ** | ** | ** | ** | ** |
| C | ** | ** | ** | ** | ** |
| AXB | NS | ** | ** | ** | ** |
| AXC | ** | ** | NS | ** | * |
| BXC | ** | ** | ** | ** | ** |
| BXC | * | ** | ** | ** | ** |
| AXBXC | ** | ** | ** | ** | ** |

Values are the means of three replicates. According to Duncan's multiple range test, values followed by the same letter within a column are not significantly different at the 0.01% level of probability. ** Significant at the 1% level, NS: statistically non-significant, * Significant at the 5% level. Factors: A1 (control with distilled water); A2 (microbial consortia); B1 (control Soil); B2 (2 Ec); B3 (5 Ec); B4 (8 Ec); C1 (O days); C2 (15 days after transplanting); C3 (35 days after transplanting).

**Table 3.** Effects of microbial consortia on the biomass of tomato plants at different salinity levels.

| Treatment | Dry Biomass Root (g) | | Dry Biomass Shoot (g) | | Dry Biomass Leaves (g) | |
|---|---|---|---|---|---|---|
| | 15 days | 35 days | 15 days | 35 days | 15 days | 35 days |
| A1B1 | 0.58 [h] | 0.69 [h] | 0.48 [h] | 0.99 [h] | 0.85 [h] | 1.26 [h] |
| A1B2 | 0.60 [e] | 0.69 [e] | 0.45 [e] | 1.22 [e] | 0.68 [e] | 1.34 [e] |
| A1B3 | 0.48 [c] | 0.55 [c] | 0.36 [c] | 0.84 [c] | 0.58 [c] | 1.13 [c] |
| A1B4 | 0.25 [a] | 0.44 [a] | 0.18 [a] | 0.44 [a] | 0.27 [a] | 0.63 [a] |
| A2B1 | 0.77 [g] | 0.75 [g] | 0.51 [g] | 1.22 [g] | 0.91 [g] | 1.55 [g] |
| A2B2 | 0.97 [f] | 0.94 [f] | 0.70 [f] | 1.78 [f] | 1.20 [f] | 2.03 [f] |
| A2B3 | 0.62 [d] | 0.73 [d] | 0.53 [d] | 1.32 [d] | 0.85 [d] | 1.36 [d] |
| A2B4 | 0.52 [b] | 0.58 [b] | 0.36 [b] | 0.89 [b] | 0.55 [b] | 1.13 [b] |
| A | ** | ** | ** | ** | ** | ** |
| B | ** | ** | ** | ** | ** | ** |
| AXB | NS | NS | NS | NS | NS | NS |

Values are the means of three replicates. According to Duncan's multiple range test, values followed by the same letter within a column are not significantly different at the 0.01% level of probability. ** Significant at the 1% level, NS: statistically non-significant.

The other growth parameters viz., root length, shoot length, the number of leaves, and the number of secondary roots showed a noticeable reduction in the control uninoculated plants (A1B1 to A1B3) (Figure 1a,c,e). A similar trend of decline was also observed with these parameters as mentioned for dry mass. There was a significant reduction in root length, shoot length, the number of leaves, and the number of secondary roots by 72.98% (3.0 cm), 62.95% (7.10 cm), 59.34% (12.33), and 61.48% (19.00), respectively, at 8 dS/m after 15 days (Table 4). After inoculation of consortia to the plants grown at 0.2 dS/m to 8.0 dS/m, the root length, shoot length, the number of leaves, and the number of secondary roots decreased by 44.08% (7.4 cm), 36.27% (13.70 cm), 32.29% (21.67), and 35.88% (36.33) at 8.0 dS/m from their 0.2 dS/m counterpart, respectively, after 15 days. There was a marginal reduction in the value at 8.0 dS/m (Figure 1f) after 35 days of inoculation compared to 0.2 dS/m (Table 4) (Figure 1b).

**Table 4.** Effects of microbial consortia on the growth-related parameters at different salinity levels.

| Treatment | Root Length (cm) | | Shoot Length (cm) | | Number of Leaves | | No. of Secondary Roots | | Chlorophyll Content (mg/g) | |
|---|---|---|---|---|---|---|---|---|---|---|
| | 15 DAS | 35 DAS | 15 DAS | 35 DAS | 15 DAS | 35 DAS | 15 DAS | 35 DAS | 15 DAS | 35 DAS |
| A1B1 | 11.2 [h] | 24.8 [h] | 19.17 [h] | 55.78 [h] | 30.33 [h] | 49.33 [h] | 49.33 [h] | 73.33 [h] | 3.28 | 3.90 |
| A1B2 | 9.6 [e] | 24.9 [e] | 19.70 [e] | 51.97 [e] | 28.67 [e] | 46.00 [e] | 47.67 [e] | 66.33 [e] | 3.60 | 4.64 |
| A1B3 | 6.9 [c] | 18.2 [c] | 14.13 [c] | 48.91 [c] | 21.00 [c] | 36.33 [c] | 39.33 [c] | 48.67 [c] | 6.18 | 7.71 |
| A1B4 | 3.0 [a] | 9.7 [a] | 7.10 [a] | 24.01 [a] | 12.33 [a] | 19.00 [a] | 19.00 [a] | 20.33 [a] | 6.96 | 7.35 |
| A2B1 | 13.3 [g] | 27.9 [g] | 21.50 [g] | 60.09 [g] | 32.00 [g] | 56.33 [g] | 56.67 [g] | 90.00 [g] | 5.70 | 4.92 |
| A2B2 | 13.4 [f] | 40.5 [f] | 31.20 [f] | 76.12 [f] | 45.00 [f] | 76.33 [f] | 70.67 [f] | 93.33 [f] | 6.15 | 5.61 |
| A2B3 | 10.8 [d] | 31.8 [d] | 23.60 [d] | 63.43 [d] | 30.67 [d] | 51.67 [d] | 51.67 [d] | 75.67 [d] | 5.58 | 5.67 |
| A2B4 | 7.4 [b] | 17.8 [b] | 13.70 [b] | 41.75 [b] | 21.67 [b] | 34.67 [b] | 36.33 [b] | 49.67 [b] | 2.68 | 1.27 |
| A | ** | ** | ** | ** | ** | ** | ** | ** | NS | ** |
| B | ** | ** | ** | ** | ** | ** | ** | ** | ** | ** |
| AXB | NS | NS | ** | NS | ** | NS | NS | NS | ** | ** |

Values are the means of three replicates. According to Duncan's multiple range test, values followed by the same letter within a column are not significantly different at the 0.01% level of probability. ** Significant at the 1% level, NS: statistically non-significant.

The root length (A2B2, 40.5 cm), shoot length (A2B2, 76.12 cm), number leaves (A2B2, 76.33), and number of secondary roots (A2B2, 93.33) were found to be significantly higher at 3 dS/m with microbial consortia; thus, it indicates that the optimal soil Ec and amendment of microbial consortia both have a positive impact on plant growth. However, the chlorophyll content showed a significantly higher improvement of 73% than the uninoculated control. The chlorophyll content was reduced gradually by 61.49% and 74.18% after 15 days and 35 days of post inoculation, respectively, even in the presence of microbial consortia, which indicated the negative effect of excess salt on photosynthesis, and when compared to the non-inoculated plants, the microbial consortia increased the chlorophyll content. The value present in the Table 4 reflects the effects of the consortia.

### 3.2. Measurement of Na+ and K+ Uptake

The data in Table 5 represents the observations taken at 0 days, 15 days, and 35 days after inoculation. There was no correlation found between the mineral content of the plant organs in control and inoculated plants. The Na+ and K+ contents of the root were gradually increased as the days of inoculation and incubation reached 35 days (Table 5). The same trend was observed with the Na+ and K+ contents in leaf and shoot, but the shoot potassium content decreased at a higher rate as the Ec level increases, and shoot showed a higher value of K+ at 15 days of post inoculation compared to the 35 days post inoculation. The microbial consortia did not protect plants from the uptake of K and Na ($p < 0.01$). The salt uptake and their transport are related to the K/Na ratio and the K–Na selectivity (Table 5).

**Table 5.** Effects of microbial consortia on uptake of $K^+$ and $Na^+$ in leaf, stem, and root.

| Treatments | Leaf | | Stem | | Root | |
|---|---|---|---|---|---|---|
| | $Na^+$ | $K^+$ | $Na^+$ | $K^+$ | $Na^+$ | $K^+$ |
| A1B1C1 | 0.54 [o] | 1.64 [o] | 1.06 [o] | 8.91 [o] | 0.49 [o] | 0.85 [o] |
| A1B1C2 | 0.68 [o] | 1.63 [o] | 0.24 [o] | 1.40 [o] | 0.71 [o] | 0.48 [o] |
| A1B1C3 | 0.36 [o] | 1.88 [o] | 0.18 [o] | 1.38 [o] | 2.16 [o] | 4.40 [o] |
| A1B2C1 | 1.04 [l] | 1.30 [l] | 1.18 [l] | 7.65 [l] | 0.82 [l] | 1.13 [l] |
| A1B2C2 | 2.42 [j] | 1.46 [j] | 3.62 [j] | 7.40 [j] | 1.83 [j] | 1.36 [j] |
| A1B2C3 | 1.28 [hi] | 0.52 [hi] | 0.95 [hi] | 0.80 [hi] | 3.49 [hi] | 1.04 [hi] |
| A1B3C1 | 1.27 [g] | 1.59 [g] | 3.87 [g] | 14.61 [g] | 0.38 [g] | 1.01 [g] |
| A1B3C2 | 1.85 [f] | 1.36 [f] | 4.81 [f] | 9.52 [f] | 1.52 [f] | 0.90 [f] |
| A1B3C3 | 2.07 [f] | 1.32 [f] | 1.17 [f] | 1.12 [f] | 2.74 [f] | 1.64 [f] |
| A1B4C1 | 0.61 [c] | 1.23 [c] | 1.11 [c] | 8.45 [c] | 0.65 [c] | 1.08 [c] |
| A1B4C2 | 1.06 [b] | 0.96 [b] | 3.65 [b] | 9.30 [b] | 1.94 [b] | 2.18 [b] |
| A1B4C3 | 0.65 [a] | 2.26 [a] | 0.36 [a] | 1.10 [a] | 0.90 [a] | 1.34 [a] |
| A2B1C1 | 1.21 [n] | 1.79 [n] | 0.28 [n] | 1.38 [n] | 0.47 [n] | 0.57 [n] |
| A2B1C2 | 0.66 [n] | 1.55 [n] | 0.27 [n] | 1.47 [n] | 0.79 [n] | 0.88 [n] |
| A2B1C3 | 0.36 [n] | 1.68 [n] | 0.29 [n] | 1.49 [n] | 1.06 [n] | 1.24 [n] |
| A2B2C1 | 1.24 [k] | 1.53 [k] | 1.63 [l] | 8.91 [l] | 0.77 [k] | 0.69 [k] |
| A2B2C2 | 1.17 [l] | 1.55 [l] | 3.47 [l] | 9.87 [l] | 1.27 [l] | 0.84 [l] |
| A2B2C3 | 0.97 [m] | 1.73 [m] | 0.60 [m] | 1.15 [m] | 2.98 [m] | 2.45 [m] |
| A2B3C1 | 1.52 [h] | 1.39 [h] | 2.35 [h] | 8.93 [h] | 0.82 [h] | 1.14 [h] |
| A2B3C2 | 1.56 [i] | 1.63 [i] | 3.24 [i] | 7.56 [i] | 1.73 [i] | 1.13 [i] |
| A2B3C3 | 1.98 [j] | 1.56 [j] | 1.09 [j] | 0.77 [j] | 3.29 [j] | 2.04 [j] |
| A2B4C1 | 1.63 [c] | 1.43 [b] | 1.64 [b] | 8.37 [b] | 0.79 [b] | 0.53 [b] |
| A2B4C2 | 3.43 [d] | 1.71 [d] | 6.38 [d] | 11.17 [d] | 2.29 [d] | 1.18 [d] |
| A2B4C3 | 3.07 [e] | 1.69 [e] | 1.11 [e] | 1.22 [e] | 3.71 [e] | 2.36 [e] |
| A | ** | ** | NS | ** | ** | ** |
| B | ** | ** | ** | ** | ** | ** |
| C | ** | NS | ** | ** | ** | ** |
| AXB | ** | ** | ** | ** | ** | ** |
| AXC | ** | NS | ** | ** | ** | * |
| BXC | ** | ** | ** | ** | ** | ** |
| BXC | ** | ** | ** | ** | ** | ** |
| AXBXC | ** | ** | NS | ** | ** | ** |

Values are the means of three replicates. According to Duncan's multiple range test, values followed by the same letter within a column are not significantly different at the 0.01% level of probability. ** Significant at the 1% level, NS: statistically non-significant, * Significant at the 5% level.

There was a 30 to 60% reduction in the K/Na ratio in leaf, stem, and root in the uninoculated plants with increasing Ec levels at 15 days, and this indicates the increase in potassium levels by the plant. Evaluation of K–Na selectivity reflected no significant changes in any of the plant parts in the uninoculated and inoculated plants. The leaf, shoot, and root of the inoculated control plants had a significantly greater K–Na selectivity than the control plants under the non-inoculated conditions (Table 6). The inoculated and uninoculated plants showed no sign of differences in the K–Na selectivity at any Ec level understudy as the Ec level goes to 8.0 dS/m; the microbial consortia did not influence the K–Na selectivity (Table 6).

**Table 6.** Effect of microbial consortia on the $K^+/Na^+$ ratio and $K^+$–$Na^+$ selectivity in root, stem, and leaf at different salinity levels.

| Treatments | Soil | Leaf | Stem | Root | Leaf | Stem | Root |
|---|---|---|---|---|---|---|---|
| | $K^+/Na^+$ | $K^+/Na^+$ | $K^+/Na^+$ | $K^+/Na^+$ | $K^+$–$Na^+$ Selectivity | $K^+$–$Na^+$ Selectivity | $K^+$–$Na^+$ Selectivity |
| A1B1C1 | 0.33 | 3.04 | 8.41 | 1.73 | 9.24 | 25.57 | 5.28 |
| A1B1C2 | 0.26 | 2.40 | 5.83 | 0.68 | 9.38 | 22.83 | 2.65 |
| A1B1C3 | 0.27 | 5.22 | 7.67 | 2.04 | 19.43 | 28.52 | 7.58 |
| A1B2C1 | 0.12 | 1.25 | 6.48 | 1.38 | 10.60 | 54.95 | 11.68 |
| A1B2C2 | 0.11 | 0.60 | 2.04 | 0.74 | 5.39 | 18.25 | 6.63 |
| A1B2C3 | 0.15 | 0.41 | 0.84 | 0.30 | 2.66 | 5.51 | 1.95 |
| A1B3C1 | 0.14 | 1.25 | 3.78 | 2.66 | 8.98 | 27.09 | 19.07 |
| A1B3C2 | 0.19 | 0.74 | 1.98 | 0.59 | 3.96 | 10.66 | 3.19 |
| A1B3C3 | 0.20 | 0.64 | 0.96 | 0.60 | 3.12 | 4.69 | 2.93 |
| A1B4C1 | 0.07 | 2.02 | 7.61 | 1.66 | 29.87 | 112.77 | 24.61 |
| A1B4C2 | 0.11 | 0.91 | 2.55 | 1.12 | 8.58 | 24.14 | 10.65 |
| A1B4C3 | 0.10 | 3.48 | 3.06 | 1.49 | 33.50 | 29.44 | 14.35 |
| A2B1C1 | 0.27 | 1.48 | 4.93 | 1.21 | 5.41 | 18.02 | 4.43 |
| A2B1C2 | 0.32 | 2.35 | 5.44 | 1.11 | 7.26 | 16.84 | 3.44 |
| A2B1C3 | 0.16 | 4.67 | 5.14 | 1.17 | 29.24 | 32.19 | 7.33 |
| A2B2C1 | 0.10 | 1.23 | 5.47 | 0.90 | 12.18 | 53.95 | 8.84 |
| A2B2C2 | 0.13 | 1.32 | 2.84 | 0.66 | 10.07 | 21.63 | 5.03 |
| A2B2C3 | 0.23 | 1.78 | 1.92 | 0.82 | 7.60 | 8.17 | 3.50 |
| A2B3C1 | 0.08 | 0.91 | 3.80 | 1.39 | 11.94 | 49.62 | 18.15 |
| A2B3C2 | 0.15 | 1.04 | 2.33 | 0.65 | 7.06 | 15.76 | 4.41 |
| A2B3C3 | 0.17 | 0.79 | 0.71 | 0.62 | 4.73 | 4.24 | 3.72 |
| A2B4C1 | 0.11 | 0.88 | 5.10 | 0.67 | 7.79 | 45.29 | 5.95 |
| A2B4C2 | 0.13 | 0.50 | 1.75 | 0.52 | 3.78 | 13.26 | 3.90 |
| A2B4C3 | 0.17 | 0.55 | 1.10 | 0.64 | 3.17 | 6.34 | 3.67 |

$K^+$–$Na^+$ selectivity is the $K^+/Na^+$ ratio of the plant tissues divided by the $K^+/Na^+$ ratio of the soil used.

### 3.3. Estimation of Soil Soluble Cations and Soil Salinity

The data shown in Table 2 indicated that the soil Ec and cation exchange capacity of the soil decreased marginally as the salinity stress increased from 0.2 Ec to 8.0 Ec (Table 2). When soil Ec increases, and there was no change in pH while consortia inoculated the soil, Na content was gradually decreased from the respective control at a particular Ec. A similar pattern for $Na^+$ was observed with the uninoculated plants also. On the other hand, the $K^+$ ion concentration was found to increase or remain constant in all the plants under study.

## 4. Discussion

### 4.1. Plant Growth-Related Parameters

Saline soil contains high concentration of ion in the soil solution, which affects the soil's structure and water holding capacity. Repercussion in the value of nutrients, amount of organic matter, and cultivation is observed with the change in salinity value. Increasing soil salinity and saline irrigation are the most limiting factors for cultivating horticulture crops, especially tomatoes. In India, South Gujarat is the most prominent area for cultivating vegetables and is famous for its seashore area. Most of the vegetable crops have a critical value of tolerance to salt of 2 dS/m [26]. The ratio between the dry matter of the root and the epigeous part of the plant is usually observed to be constant. As observed, the higher Ec level of the soil reduced the plant growth-related parameters. Higher levels of $Na^+$ in the plants increases the osmotic potential which negatively affects the water uptake [27]. Soil salinity is a global menace to plant growth and development causing significant economic losses to tomatoes and other crops [27–29]. Increased level of salt creates water deficit conditions and prevents water uptake from soil that ultimately disturbs the osmotic balance in the affected plants [30]. The water deficiency also exerts oxidative stress and

ionic imbalance. Various plants withstand salinity conditions [31] through appropriate responses [32].

Kusale et al. [12] reported a significant reduction in root length by 32.1%, shoot height by 19.7% and 31.1% reduction in chlorophyll content in wheat seedling after 45 days growth under saline soil. They found that the inoculation of halotolerant *Klebsiella variicola* not only reverted this negative impact but also significantly improved these plant growth parameters. Improvement in plant growth parameters viz., roots, shoots, and chlorophyll are a well-known feature of many PGPR [3].

The inoculation of microbial consortia could alleviate the stress factor by providing necessary substances for their growth. The microbial amendment in the soil at rhizosphere provides hormones, siderophore, converts complex organic matters to readily available forms, and assists in the transport and uptake of molecules. These beneficial roles of microbes help the plants to tolerate various abiotic stress conditions. The length, growth, and dimension of plants are dependent on the processes of assimilation and partitioning of various carbohydrates. This ratio is observed to change only when the plant is encountered with any stressful condition. Previously, studies have proved that an increase in salinity in tomato plants can reduce leaf and stem dry matter, and root dry matter considerably [33–36]. The reduction in the primary root length and secondary root will lead to lesser absorption of nutrients which could attribute to a lower shoot, leaf dryness, and fresh biomass [33–36].

Halophilic rhizobacteria decreases the salt level in soil, protects the plant from osmotic effects, and thus, helps the growth of the plant under salinity stress while ameliorating soil salinity [37,38]. Mayak et al. [39] reported significant improvement in the fresh and dry weights of tomato seedlings at high salt concentration following the inoculation of *Achromobacter piechaudii*. Tank and Saraf [40] also reported that halophilic *Pseudomonas stutzeri* improved salt tolerance in tomato plants. Van et al. [41] found that *A. chroococcum 76A* inoculation under both moderate (50 mM NaCl) and severe (100 mM NaCl) salt stresses improved plant growth parameters, such as shoot dry weight, fresh fruit weight, and fruit numbers per plant compared to the uninoculated plant. Halophilic PGPR have proven to be beneficial for the cultivation of vegetables under salinity stress [42].

These varied yet similar works have reported, with respect to the increasing salinity tolerance of tomatoes, an increase in the fresh and dry weight of plant parts by using *Achromobacter piechaudii* [31], *P. stutzeri* [29], *A. chroococcum 76A* [32]. Abd El-Azeem et al. [43] reported a negative effect of salinity on eggplant growth and yield. PGPR that produce IAA, siderophore, and solubilize phosphate overcame stress on eggplants by increasing the root dry weight and yield [43]. The ACC deaminase, which could reduce ethylene production by cleaving its precursor and phytohormones, could be the potential mechanism for salt tolerance in plants by the microbes [44,45]. There may be changes in root morphology, the number of secondary roots, or root length due to the application of microbial consortia [46]. The present results are supported by the observation that inoculation of *Bacillus subtilis EY2*, *Bacillus atrophaeus EY6*, and *Bacillus sphaericus GC subgroup B EY30* showed a substantial increase in the fresh weight and dry weight of squash plants inoculated with consortia compared to the control plants in the presence of salinity [47]. Inoculation of PGPR in cucumbers resulted in higher plant growth parameters, more soluble sugar, and total chlorophyll content under salt stress [48]. Halotolerant *Bacillus* consortia assisted potato tubers in salt stress conditions by producing phytohormones and antioxidant enzymes [43].

Moreover, there was a reduction in photosynthesis levels under high salinity due to the lower photosynthetic area of the leaf. Therefore, salt-stressed plants have a lower amount of accumulated carbon to be used for growth. In the present study, there was a reduction in chlorophyll content with an increase in electric conductivity. The microbial consortia helped the plant resist the stress until Ec 5 dS/m, and a further increase in salt stress led to a drastic reduction in chlorophyll content. The results were in accordance with a previous study carried out by Molazem et al. [49], in which there was a statistically significant decrease in both chlorophyll a and chlorophyll b during higher concentrations of salts compared to the control. Accumulation of ions and functional disorder in stomatal

opening and closing can be a possible reason for the depletion of the total amount of chlorophyll content under salt stress [50,51]. The rapid maturation of leaves under salt conditions can be another reason for the decrease in chlorophyll content. A few studies found that a decrease in chlorophyll content under salinity stress can be a reason for sensitive genotypes compared to halotolerant cultivar [52].

### 4.2. Na$^+$ and K$^+$ Uptake

K/Na ratio was used to determine the degree of salt tolerance in eggplants, and when there was an increase in salinity, there was a significant decrease in the K/Na ratio in the shoot, as mentioned in the previous study [46,53]. Whereas, in the present study, there was a constant decrease in the K/Na ratio and K–Na selectivity of the plants inoculated with consortia compared to the control on their respective Ec from 0 days to 35 days after inoculation. There was a higher K/Na ratio observed in the shoot as compared to the leaf and root. This lower K/Na ratio in the root can also be due to a change in root anatomy. Plants under salt stress might have thicker cell walls of root cells, and they are often convoluted. Salts promote the suberisation of the hypodermis and endodermis in woody tree roots, resulting in a well-developed casparian strip closer to the root apex, different from that found in non-salinized roots. There is a decrease in root length with an increase in secondary roots to withstand the adverse condition. The secondary roots and increased root system are formed due to the interaction with consortia storing excess Na$^+$ and do not allow the movement of excessive Na$^+$ from the root to shoot. Therefore, the acquisition of mineral nutrients and water and the tolerance of the inoculated plants to the presence of potentially toxic levels of Na+ in the soil solution may depend on the continued growth of the adventitious roots. The K/Na ratio and its content in leaves are usually accumulating fewer Na$^+$ ions in the transpiring regions. Further, with the increase in salt stress, there is also a change in the leaf's anatomy. The microbial inoculation reduces the Na accumulation of the leaf and relieves the salt stress exerted on radish and rice plants [52–56].

According to various studies, it is found that whenever a plant is subjected to salt stress, higher Na$^+$ content is found in the stem, whereas less accumulation of Na$^+$ ion is found in leaves. Here in our results, we can see an increase in Na+ in leaves while the amount of K$^+$ increased or remained almost unchanged in the inoculated plants. While there was a reduction of Na$^+$ content in stem from 0 to 35 days after inoculation, there was a slight increase in K$^+$ ions observed even at a higher Ec (8 dS/m). This shows that there was an exclusion of Na$^+$ with the assimilation of K$^+$ ions in the stem. The roots system is highly prone to salinity stress, and though there was a steady increase in the Na$^+$ content still, there was an increase in the K$^+$ content. Although, there was an influx of sodium, the potassium uptake steadily increased. K$^+$/Na$^+$ ratio determines the salt stress in the plants. K$^+$–Na$^+$ selectivity shows the uptake of Na$^+$ and K$^+$ ions from soil to the various parts of the plants. The reduction in K$^+$–Na$^+$ selectivity is more in the inoculated plants as compared in the uninoculated. Therefore, the K–Na selectively of the shoot tissues was significantly upregulated when compared to the control after inoculation with microbial consortia in the salinity stress. This can be a primary effort to attribute restriction of Na+ uptake by tomato roots and translocate it from root to shoot [53].

Kuslae et al. [53] reported that inoculation of halotolerant *K. variicola* SURYA6 significantly improved N, P, Na, K, and Mg contents in wheat and maize under salinity. This isolate resulted in a 35.1 to 89.9% increase in the mineral content compared to the uninoculated plants under salinity. They further reported a significant improvement in the level of these minerals under salinity conditions compared to normal soil.

Under salinity stress, plants accumulate proline and soluble sugars and ions to maintain osmotic adjustment [53]. Nia et al. [54] reported a higher accumulation of Na$^+$ contents in leaves and shoots, respectively, in plants inoculated with halophilic *Azospirillum* vis-à-vis the control plants and plants inoculated with non-halophilic *Azospirillum*. They claimed the better growth of the wheat plant that received *Azospirillum* inoculation under salinity

stress. Nia et al. [54] found that the improved growth of wheat plants under salt stress was due to an increase in photosynthetic pigments and more accumulation of solutes.

Moreover, the higher K/Na ratio in the shoots of inoculated tomato plants compared with the uninoculated plants indicated that the selective uptake of $K^+$ had occurred, which seems to be one of the processes involved in the tolerance of tomato plants to salinity stress. The depression of $Na^+$ uptake by $K^+$ could be due to the antagonism between two cations. Salt tolerance in many crop plants depends on the efficiency of the root system, which can regulate the excess of $Na^+$ and chlorine or sulfate ions to reach the shoot [53,55]. All tested strains significantly upregulated tomato plant root systems supporting the hypothesis that microbial consortium is an efficient method for alleviating salt stress in tomato plants, which gave a better response in terms of growth at tolerable Ec levels than their respective control.

## 5. Conclusions

The result obtained in the current paper revealed that the investigated bacterial consortia have PGP traits that can alleviate salt stress. Thus, the inoculated halophilic and halotolerant PGPR consortia mentioned in the results enhanced the tolerance limit of the tomato plants from 5 dS/m to 8 dS/m. This consortium enabled the plants to survive properly at electric conductivity higher than its normal tolerance level. Taken together, the inoculation of microbial consortia showed promising salt tolerance in tomato plants. The results suggested that tolerance mechanisms reside in the capacity of $Na^+$ accumulation in stem tissue, resulting in reduced $Na^+$ transport to the leaves at the initial stage of plant development. Higher dry matter accumulation and a sustainable amount of chlorophyll levels were observed in inoculated plants compared to control after 35 days of transplanting. Thus, this study underlines the commercial potential of these selected PGPR isolates for sustainable tomato cultivation under salt stress.

**Author Contributions:** Writing—original draft, C.K.; writing—review and editing, R.Z.S., H.A.E.E., A.S., K.A. and A.T.K.Z.; Formal Analysis, H.V., D.S., R.A.M.; methodology, H.V., D.S.; conceptualization and supervision, N.P.; fund acquisition, A.M.E. All authors have read and agreed to the published version of the manuscript.

**Funding:** This work was funded by The Researchers Supporting Project Number (RSP/2021/367), King Saud University, Riyadh, Saudi Arabia, Fundamental Research Grant Scheme (FRGS/1/2020/STG01/UPM/02/6 vote number 5540394) supported by Malaysian Ministry of Higher Education, Malaysia, Universiti Teknologi Malaysia (UTM), Allcosmos Industries Sdn. Bhd. and Arif Efektiv Sdn. Bhd. with project No. R.J130000.7344.4B200 and R.J130000.7609.4C187 and The Navsari Agricultural University, Navsari, India.

**Institutional Review Board Statement:** Not applicable.

**Informed Consent Statement:** Not applicable.

**Data Availability Statement:** Not applicable.

**Acknowledgments:** The authors extend their appreciation to the Researchers Supporting Project Number (RSP/2021/367), King Saud University, Riyadh, Saudi Arabia, Ministry of Higher Education, Malaysia, RMC, Universiti Teknologi Malaysia (UTM), Allcosmos Industries Sdn. Bhd. Arif Efektiv Sdn. Bhd. and The Navsari Agricultural University Navsari, India for financial support and facilities.

**Conflicts of Interest:** The authors declare that they have no conflict of interest.

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
