# Peer review of "Halotolerant Microbial Consortia for Sustainable Mitigation of Salinity Stress, Growth Promotion, and Mineral Uptake in Tomato Plants and Soil Nutrient Enrichment"

_sustainability, doi:10.3390/su13158369_

Round 1

Reviewer 1 Report

Introduction

We recommend the authors to detail this paragraph.

Pag.2, row 31 – repetition: that, that

Reviewer 1 Report

The authors are thankful to the reviewer for excellent reviewing of the and for endorsing the MSS

Comments

  • Introduction

We recommend the authors to detail this paragraph.

Authors response: The introduction is now written in detail. Information on salinity stress, its effects and various measures are now added. Line No. 1-90

  • 2, row 31 – repetition: that, that

Authors response: Repeating word that that is now deleted. Line No. 90

Reviewer 2 Report

This is the revision of the manuscript number Sustainability-1260270 Title: “Halotolerant microbial consortia for sustainable mitigation of salinity stress and growth promotion, and mineral uptake in tomato plant and soil nutrient enrichment”, proposed by Jillian Chen and colleagues for consideration for publication in Sustainability.

The manuscript raises a novel issue and attempts to address an increasingly problematic question and that is the loss of quality of food-producing soils, the proposed ideas in the manuscript seem excellent to me. . However, some suggestions should be considered to improve the manuscript.

Material and methods:

Because it has not been proposed to use different varieties of tomatoes?, so a genetic improvement could be achieved apart from a management strategy such as those proposed in this manuscript.

Because they simply reached 108 UFC/ml?

In the process that the plants emerge and are placed in the greenhouse at 70% humidity, it is not possible that in the greenhouse they are placed at 300º C temperature, I suppose it is a mistake.

Was any measurement of nutrients made to seawater? it may contain some elements that block the assimilation of nutrients in the environment where seawater has been applied.

How was the irrigation applied to the pots where the tomatoes were planted quantified? the application of continuous irrigation promotes the reduction of salts to lower areas of the pots, obtaining in the pot a low availability of salts by the roots of the tomato.

Results and discussion:

Excellent, a very good discussion.

General comments:

The author reports that the soils raised in pots where the tomato is grown with the addition of sterilized sea water, it would be interesting to make correlations with experiments tested in field soils, where there are conditions that promote more variables.

Author Response

 Response to 2nd Reviewer’s comments:-

  1. Comments: Because it has not been proposed to use different varieties of tomatoes? So a genetic improvement could be achieved apart from a management strategy such as those proposed in this manuscript.
  2. Because they reached 108 UFC/ml?
  • Response: The farmer requires technology at a cheaper rate with less environmental impact. The genetic modification of plants requires many safety measures, yet the GEAC committee approves only one crop in India for commercialization. The present aim was to amend the soil with consortia with PGPR traits; thus, soil fertility does not compromise, and plant growth could be achieved in saline soil. The organism load was 108 Colony-forming Units per ml was used.
  1. Comments: In the process that the plants emerge and are placed in the greenhouse at 70% humidity, it is not possible that they are placed at 300º C temperature in the greenhouse. I suppose it is a mistake.
  • Response:- Plants were grown in plug tray at 30 º 300 ºC is a typological mistake now corrected as 30 ºC (Page 4 and line 107). They were transferred to pot in normal sunlight. The experiment was performed with different levels of salinity in the soil (Ec Electric Conductivity). We mimic the natural rhizosphere condition available in the saline rhizosphere in terms of the Ec.
  1. Comments:- Was any measurement of nutrients made to seawater? It may contain elements that block the assimilation of nutrients in the environment where seawater has been applied.
  • Response: Farmers’ uses farm water for fertigation purpose and due to seashore in nearby; their sources of water and land get affected by the salinity. The vegetables are the most sensitive to the saline environment. The analysis of seawater was not included as the aim of the present study was to evaluate the role of consortia in mitigation of salt stress with natural conditioned developed by applying seawater in the soil before initiation of the experiment.
  1. Comments: How was the irrigation applied to the pots where the tomatoes were planted quantified? The application of continuous irrigation promotes the reduction of salts to lower areas of the pots, obtaining a low availability of salts by the roots of the tomato.
  • Response:- Irrigation was not quantified as experiments were set up to see the plant's growth during interactions with salinity stress and microbes. We applied irrigation at a regular interval of three days, and the tomato roots do not go in-depth, and the pot was selected to achieve a homogenous condition.

Reviewer 3 Report

The current MS is designed to develop the use of microbial consortia to manage soil salinity and improve plant growth and development in tomato. The overall goal of this MS looks promising but it need to addressed number of  things before making any final conclusion. here is my step by step comments:

  1. the very first sentence of introduction is very vague "Agriculture serves as the basic aid to fulfill the nutrition requirement throughout the 
    world." If you think on broader sense it is much mire than that.
  2. The projected global population in 2050 is much more than 2050. Please cite proper direct reference for this too. 
  3. The introduction is very short and not updated on salinity stress and various measures being taken currently. 
  4. The and objective of this MS is not very clear. 
  5. Tomato seeds (Solanum Lycopersicum L. cv. Arka Vikas): Is this variety tolerant or susceptible to salinity? It will be nice this has been done with both tolerant and susceptible variety. 
  6. No information on how this 6 microbial cultures were selected initially. 
  7. Is these all microbes safe to human and have no any human pathogenicity known? Please confirm this. This is very critical.
  8. Treatment: This is not very clear or not explained at all. Why distilled water is used as source of irrigation?  What is the difference between treatment A1 and B1 treatments?
  9. "The plants were allowed to grow in the natural light and temperature". What was the temperature during the experimental period? Did you check the light intensity?
  10. Since these microbes were also selected on the basis of siderophore activity, It will be good to see the micro nutrient (Fe, Zn, Mn, and Cu) in different tissues. 
  11. Results: There is lot of scope to improve the results section. Currently it is written in a very generic format. Please refer any current paper like this to see how the results sections is organized. 
  12. Based on results, the discussion and conclusion will changed dramatically.  
  13. Please check the spelling error. 

Author Response

Response to 3rd Reviewers comments

  1. Comments:- the very first sentence of the introduction is very vague "Agriculture serves as the basic aid to fulfill the nutrition requirement throughout the world." If you think in a broader sense, it is much more than that.

Response:- Introduction has been modified from line number 1 to 90. This line has been removed.

  1. Comments:-The projected global population in 2050 is much more than 2050. Please cite proper direct reference for this too. 

Response:- These lines have been removed.

  1. Comments:- The introduction is very short and not updated on salinity stress and various measures being taken currently. 

Response:- The introduction is now written in detail. Information on salinity stress, its effects, and various measures are now added. Line No. 1-90

  1. Comments:-The and objective of this MS is not very clear. 

Response:-The objectives have been revised for clarity. Line number 90-93.

  1. Comments:-Tomato seeds (Solanum Lycopersicum L. cv. Arka Vikas): Is this variety tolerant or susceptible to salinity? It will be nice this has been done with both tolerant and susceptible variety. 

Response:- The commercially grown tomatoes have a threshold level up to 2.5 dSm-1, (Singh J et al., 2012). This variety is predominantly grown by the farmer in this region, and this variety faces yield loss due to diverse salinity levels. There is a need to amend the soil, supporting the plant growth above 7-8 Ec. The susceptible and tolerant plant analysis would have different objectives, and we aimed to notice the role of halotolerant consortia on tomato growth promotion under high Ec-containing soil.   

  1. Comments:-No information on how this 6 microbial cultures were selected initially. 

Response:- Table 1 explains the characteristics of the microbes to be used as PGPR. The organisms were isolated from the saline rhizosphere and characterize in the laboratory. The detailed characteristics of organisms are presented in the different papers, and those are under review.

  1. Comments:-Is these all microbes safe to humans and have no human pathogenicity known? Please confirm this. This is very critical.

Response:- The majority of the organisms used in current experiments belongs to GRAS family bacteria (Generally recognized as Safe) identified by US FDA and reported by Denner and Gillanders, 1996 as well as reviewed by Aloo et al., 2019. Moreover, they have wide acceptance in commercial preparation for other mentioned bacteria and are reported by many previous scientific findings.

  1. Comments:-Treatment: This is not very clear or not explained at all. Why is distilled water used as source of irrigation?  What is the difference between treatment A1 and B1 treatments?

Response:- Line number 109-113 explained the preparation of pot with different Ec levels. The plant requires water at regular interval, and normal tap water also contains some amount of salts. To reduce the error during the experiment and get optimum results of plant growth-related parameters resulting from consortia. The A1 was only controlled soil without consortia, and A2 was with consortia, while B1 to B4 were four Ec level pots. A1B1 was considered to be a control plant with normal Ec of the soil with distilled water. The A1B2 was the controlled pot with the amendment of consortia.  This gave an idea about the effect of consortia in the nonsaline soil as a control measure during experiments.

  1. Comments:-"The plants were allowed to grow in the natural light and temperature." What was the temperature during the experimental period? Did you check the light intensity?

Response:- The plants were kept in normal sunlight along with control pots, and light intensity was not measured. All plants under experiments involving pot were receiving homogeneous environments to nullify any environmental impact.

  1. Comments:-Since these microbes were also selected based on siderophore activity, It will be good to see the micronutrient (Fe, Zn, Mn, and Cu) in different tissues. 

Response:- The micronutrient was not analyzed as salinity involved Na, K majorly, Ca, and Mg.

  1. Comments:- Results: There is a lot of scopes to improve the results section. Currently, it is written in a very generic format. Please refer to any current paper like this to see how the results section is organized. 
  2. Comments:-Based on results, the discussion and conclusion will changed dramatically.  
  3. Comments:-Please check the spelling error. 

Response:- The spelling has been checked deliberately.

Round 2

Reviewer 3 Report

This revised MS stills needs lot of improvement.  Seems that only cosmetic changes has been made. 

  1. The results and discussion needs more attention. As I mentioned before, The results and discussion is described in a very generic format.  Try to improve this section, otherwise it makes reading really tough.
  2.  Table 3: The dry biomass between different samples looks very good, in some cases it is about double than the control treatment. Please provide the phenotypic pictures of these all samples at different time-point of different treatment conditions.  This table needs to validated with qualitative figures. 

Author Response

Response to Reviewer 3 Round 2 Report

  1. The results and discussion needs more attention. As I mentioned before, the results and discussion is described in a very generic format.  Try to improve this section, otherwise it makes reading really tough.

Response:- The results and discussion section has been improved and made more simplified. Line number 157-181, 190-199 of Results and Line No. 259-281, 287-295, 355-365 of discussion.

  1. Table 3: The dry biomass between different samples looks very good, in some cases it is about double than the control treatment. Please provide the phenotypic pictures of these all samples at different time-point of different treatment conditions.  This table needs to validated with qualitative figures. 

Response:  The 06 images of the treatment are provided as suggested.

Round 3

Reviewer 3 Report

This MS is improved than before. 

few minor things to consider before acceptance:

  1. Figure 1 Legend is missing.
  2. Include pictures of all treatment conditions
  3. The treatment mentioned in figure 1 and in text is not matching. 
  4. In figure 1e and 1f: A1B3 is with  Ec value of 8.0 while in text (materials and methods) A1B3 is having Ec value of 5.0 only. similarly there is other mismatching too. 

Author Response

REVIEWER 3 Round 3 Report

This MS is improved than before. few minor things to consider before acceptance:

Authors response: The authors are thankful to the reviewers for an excellent review of the MSS. This Review has helped in the significant improvement of the MSS

  • Figure 1 Legend is missing.

Authors response: Legends have been added in Fig 1

  • Include pictures of all treatment conditions

Authors response: Pictures of all treatments that had significant effects  have been included

  • The treatment mentioned in figure 1 and in the text is not matching. 

Authors response: Revised. Line No. 137, 177, 200, 201

  • In Figures 1e and 1f: A1B3 is with Ec value of 8.0 while in the text (materials and methods) A1B3 is having Ec value of 5.0 only. Similarly there is other mismatching too. 
  • Authors response: In figure 1e A1B3 is now corrected as Ec value of 5.0 while in text.